biomechanics, plant science, physiology

cost-benefit analysis, pitfall trap, plant movement, prey capture, snap trap, suction trap

**Authors for correspondence:**
Ulrike Bauer
e-mail: ulrike.bauer@bristol.ac.uk
Ulrike K. Müller
e-mail: umuller@mail.fresnostate.edu
Simon Poppinga
e-mail: simon.poppinga@biologie.uni-freiburg.de

†These authors contributed equally to this work.

# Complexity and diversity of motion amplification and control strategies in motile carnivorous plant traps

Ulrike Bauer[1,†], Ulrike K. Müller[2,†] and Simon Poppinga[3,4,†]

[1]School of Biological Sciences, University of Bristol, Bristol, UK
[2]Department of Biology, California State University Fresno, Fresno, CA, USA
[3]Plant Biomechanics Group, Botanic Garden, and [4]Cluster of Excellence livMatS @ FIT—Freiburg Center for Interactive Materials and Bioinspired Technologies, University of Freiburg, Freiburg, Germany

 UB, 0000-0002-4701-793X; UKM, 0000-0001-5338-9310; SP, 0000-0001-5341-9188

Similar to animals, plants have evolved mechanisms for elastic energy storage and release to power and control rapid motion, yet both groups have been largely studied in isolation. This is exacerbated by the lack of consistent terminology and conceptual frameworks describing elastically powered motion in both groups. Iconic examples of fast movements can be found in carnivorous plants, which have become important models to study biomechanics, developmental processes, evolution and ecology. Trapping structures and processes vary considerably between different carnivorous plant groups. Using snap traps, suction traps and springboard-pitfall traps as examples, we illustrate how traps mix and match various mechanisms to power, trigger and actuate motions that contribute to prey capture, retention and digestion. We highlight a fundamental trade-off between energetic investment and movement control and discuss it in a functional-ecological context.

## 1. Introduction

'Biologists have long been attracted to (…) extremes because they provide especially clear examples from which to determine structure-function relations' [1, p. 100]. Extremes of movement are particularly prevalent in predator–prey interactions when organisms prioritize high speed and acceleration during predator strikes or escape responses, such as the claw strikes of mantis shrimp and the escape jumps of trap-jaw ants [2]. In animals, many of these ultra-fast movements are spring-driven [3–6]. Springs can be used to overcome power limits, particularly by animals small enough to benefit from the combination of springs' high power output with a small body mass [3]. Recent studies of spring-driven systems illuminate the importance of conceptual frameworks and careful terminology, replacing old concepts such as 'power amplification' [7] with new ones, such as 'latch-mediated spring actuation' [4]. This new framework uses energy flow through the elastic system to identify its parts (motor, latch, spring and actuated mass) and processes (latching, loading and launching) [4,8,9]. Storage and release of energy by springs have often been framed as a means to overcome limitations imposed by muscles' ability to produce power [7,10–12]. The new framework acknowledges additional and alternative rationales for using springs, such as energy efficiency, motion control, impedance matching between actuators and load, and thermal robustness [3,4,7,8,13–16].

The mechanisms and energetics of ultra-fast movements have largely been studied in animals; however, rapid predatory strikes are also used by many carnivorous plants. Can we apply the same principles and frameworks to plant movements? Carnivorous plants differ from animal predators in fundamental ways: (i) plants do not use muscles as motors [17], (ii) plants derive essential

nutrients rather than energy from prey ([18], but see [19]), and (iii) plants are modular organisms capable of continuously growing and replacing functional organs. A modular Bauplan not only relaxes the need for structural durability, as plants can replace where animals need to repair, but it also creates redundancy, which in turn facilitates the evolution of specializations, i.e. subsets of organs with distinct functions [20] as well as morphological plasticity—carnivorous plants vary their biomass investment in traps in response to nutrient and prey availability [21–24]. In animals with their integrated Bauplan, such evolutionary innovation [25] and plasticity are less common. Few animals vary their morphology in response to prey availability or nutritional status—for example, larvae of the salamander have two different prey-induced morphs [26], and pythons absorb energy-expensive digestive organs when they are not digesting prey [27].

Owing to their stiff cells walls, plants cannot employ contractile proteins such as the muscle fibres of animals. Instead, most plant movements (e.g. the opening and closing of stomata [28]) are powered by changes of hydrostatic pressure (turgor), driven by energy-requiring water displacement processes between cells and tissues [29]. Hydraulic actuation is rarely used by animal predators [30,31]. While animal movements are generally limited by muscle power output [32], the speed of hydraulic plant movements is ultimately limited by the rate of fluid transport across cell membranes [33]. Both animals and plants incorporate the release of elastic energy to achieve movement speeds and accelerations beyond these physiological limits.

Motile traps of carnivorous traps have traditionally been classified as 'active' as opposed to 'passive' (non-motile) traps. This terminology has been called into question [34], because it lumps together multiple processes that contribute to a prey capture event (such as prey attraction, capture, retention and digestion), and because it confounds motion with control and energetic investment. In this review, we distinguish three aspects of what has previously been subsumed in the terms 'active' and 'passive': (i) motion, (ii) the conversion of metabolic energy into movement and (iii) the relative timing of energy conversion and motion. (i) We use 'motile' (versus 'non-motile') to refer to traps that move during some part of the prey capture process. The proposed terminology for traps complements established terminology for plant movements more generally [35,36]. (ii) We use 'intrinsic' to denote that energy to trigger or power the motion is supplied by the metabolism of the plant and 'extrinsic' if the energy comes from other organisms or abiotic factors (such as wind or gravity). Powering and/or triggering motion extrinsically implies that the organism no longer entirely controls the process. (iii) We use 'synchronous' when energy is directly converted into movement, and 'asynchronous' when energy input and release are separated in time. Asynchronous traps convert metabolic energy first into potential energy, which is stored in elastic structures and released as kinetic energy at a later point. In the following, we will discuss how animal and plant predators differ in using intrinsically versus extrinsically and synchronously versus asynchronously powered mechanisms to trigger and power motion-based capture events. We then present three case studies of carnivorous plants with fast motile traps that exemplify the usefulness of distinguishing along these three axes of motion, energy and timing. These examples also illustrate how animals versus carnivorous plants address the biomechanical demands of being ambush predators. We focus on prey capture and retention because these two processes are at the heart of how 'active' versus 'passive' traps have traditionally been defined.

## 2. Distinguishing animal and plant predators along the axes of motion, energy and timing

Animal and plant predators may employ motile or non-motile mechanisms to capture prey. Movements differ with respect to the energy source (intrinsic versus extrinsic) and timing (synchronous versus asynchronous). Non-motile traps often employ adhesives or pitfall mechanisms.

### (a) Motile, intrinsically powered, synchronous

Most animal predators power fast prey capture intrinsically and synchronously—i.e. using muscle power directly. The speed of such muscle-powered movements is ultimately limited by (i) the power output rate of the muscle which is capped at approximately $300 \, W \, kg^{-1}$ and decreases as contraction speed increases [32], (ii) the time available for muscle contraction—by definition short in a fast movement—and (iii) the available acceleration distance which decreases with body size. This size effect is exacerbated by the fact that in order to reach the same absolute speed, a smaller animal will need to achieve higher *relative* speed, as expressed in body lengths per time. Unsurprisingly, it is the very small animal predators that power their strikes asynchronously. This is a noteworthy difference to plants, because the speed of hydraulic plant movements is limited by the poroelastic time, i.e. the time it takes water to be transported across tissues, resulting in an inverse relationship of speed and size [33,37]. Therefore, most fast plant movements are actuated asynchronously. Intrinsically and synchronously powered, *fast* trap movements have only been described in one carnivorous plant species, *Drosera glanduligera*, a sundew with hydraulically actuated, small 'snapping' tentacles that catapult prey into glue traps. Yet, the snapping time of roughly 75 ms is considerably longer than the theoretical poroelastic time of 16 ms [38]. The synchronously powered hydraulic movements of numerous other species of sundews (*Drosera*) and butterworts (*Pinguicula*) are much slower. In these species, movement aids prey retention rather than initial capture which is effected by sticky secretions [39–41]. Intrinsically and synchronously powered fast movements are confined to larger animals, but tend to be small scale in plants.

### (b) Motile, intrinsically powered, asynchronous

Animals and plants alike use spring-driven systems to control energy flow during prey capture events. Small animal predators and carnivorous plants with fast motile traps overwhelmingly use asynchronous capture mechanisms. Intrinsically powered muscle contractions or hydraulic forces gradually load springs over time, converting metabolic energy into elastic energy. This accumulated energy is released when prey triggers the removal of a 'latch' [3,42]. Examples for spring-powered prey capture devices are the claws of mantis shrimp and the traps of the Venus flytrap [10,43]. In animals, both the loading of the spring and the release of the latch require metabolic energy [3,4]. By contrast, only two species of carnivorous plants (*Dionaea muscipula* and *Aldrovanda vesiculosa*) are known to use

intrinsically powered latches and rely on hydraulic processes during all phases of the trapping movement [44,45]. The asynchronous suction traps of bladderworts (*Utricularia*) are intrinsically powered, but extrinsically triggered, using kinetic energy provided by the prey. Across carnivorous plants, fast trap motions rely on the asynchronous release of gradually accumulated, intrinsically supplied elastic energy.

## (c) Motile, extrinsically powered, synchronous

This mode of motion actuation is currently only known from plants. The pitcher plant *Nepenthes gracilis* lures prey to the underside of its roof-like trap lid, thereby manoeuvring it into an optimal position to exploit its potential (gravitational) energy. The actual trapping motion, a high-frequency vibration that dislodges the prey into the trap, is actuated extrinsically by the kinetic energy of falling raindrops [46]. We already mentioned bladderworts, where the energy to trigger the trap opening is provided extrinsically. Prey touch and bend trigger hairs which in turn deform the trap door, causing it to snap-buckle and open [47,48]. The reduction of metabolic costs for the predator comes at the expense of motion control: a predator that relies on extrinsic energy to power or trigger prey capture movements forfeits control over the timing of the capture event and over the energy transferred during the event. Predators that retain control over their prey capture do so at some energetic expense.

## (d) Motile, extrinsically powered, asynchronous

Interestingly, this combination does not appear to be realized in nature. Even though bladderworts use asynchronous energy release to operate their suction traps, they first need to set the trap by metabolically costly water pumping processes. The externally powered triggering is a synchronous process, where externally supplied kinetic energy is directly used to unlatch the trap door. An asynchronous trapping motion where the stored elastic energy is externally supplied has not been described to date.

## (e) Non-motile

Plant, animal and human predators also use non-motile traps. We can distinguish three types of non-motile traps based on their operating principle: glue traps, pitfall traps and 'eel traps'. Eel traps use unidirectional 'valve' gates to funnel prey into a dead end. Because no movement is involved, there are no energy requirements beyond the construction and maintenance costs of the trap. Examples are the 'flypaper' leaves of non-motile sundew species and *Drosophyllum*, the 'eel traps' of the corkscrew plant *Genlisea*, the webs of orb spiders and the pitfall traps of ant lions and pitcher plants. Many traditional traps used by human hunters also exploit similar principles, e.g. the glue sticks and mist nets used by bird hunters or the bow nets of fishermen. Although not directly employing movement, non-motile traps rely to some extent on extrinsic energy, usually the kinematic and gravitational potential energy of prey.

By comparing animal and plant predators, we found similarities and differences. Both use spring-loaded mechanisms because prey capture often requires extremely rapid movements and high accelerations [2,3]. Whereas all but the smallest animals power their motion synchronously,

plants rarely do. Only plants appear to use extrinsically powered motion.

# 3. The trapping process and its energetics for three motile traps

Using three examples, we illustrate the complexity of the prey capture process focusing on the latch-mediated spring actuation framework [4], which characterizes the energy flow through a spring-driven mechanism, in our case a prey trapping mechanism. Prey capture events can be viewed as a three-step cyclic process: (i) setting of the trap by the gradual conversion of metabolic energy into elastic energy, i.e. a motor loading a spring, which is held by a latch, (ii) triggering of latch removal and energy release, and (iii) prey capture effected by the rapid conversion of potential into kinetic energy, i.e. trap movement. A return to step (i) resets the trap so it can fire again.

## (a) Example 1: intrinsically powered triggering and motion in asynchronous snap traps

Motile snap traps (figure 1; electronic supplementary material, table S1) are found in the Venus flytrap (*D. muscipula*) and in the waterwheel plant (*A. vesiculosa*), two sister species within the Droseraceae [50]. Fast trap closure is possible because of the rapid conversion of gradually accumulated elastic energy into kinetic energy. Typically, these traps catch a single prey per strike [48,51] and can catch additional prey only after resetting. The traps are set by mechanically pre-stressing the trap lobes [49,52,53]. Triggering begins with an electrophysiological signalling cascade when mechanosensory trigger hairs on the inside of the trap are bent [52,54]. In the Venus flytrap, this initiates intrinsically powered turgor changes that cause the trap lobes to deform, initially increasing the concavity of the lobes, until they snap-buckle to become convex, thereby converting mechanical pre-stress into rapid motion [53,55,56]. Both lobes move independently of each other, which occasionally results in asynchronous snapping events [57]. Hence, each lobe of the Venus flytrap works as a separate motor, spring and latch unit (figure 1).

By contrast, the traps of the waterwheel plant structurally separate motor and spring. While the motor is still located in the trap lobes, the spring is situated in the midrib between them. Upon triggering, energetically costly turgor changes in motor zones flanking the midrib [58–60] initiate the release of pre-stress [49]. The midrib flexes downwards and the trap shuts [61]. In contrast to the Venus flytrap, the lobes do not change their curvature during this process, and their movement is mechanically coupled via the midrib. As a result, the trap forms a single kinematic element and the lobes always move in synchrony [45].

After shutting, both traps further narrow and form a tightly sealed 'stomach'. The waterwheel plant achieves this via hydraulics and the continued release of elastic energy from the midrib, whereas the Venus flytrap uses metabolically powered hydraulic processes alone [49,58,62]. The traps of both species can snap repeatedly and re-open by means of energy-requiring growth processes [58,63]. In summary, snap traps convert energy during all steps of the

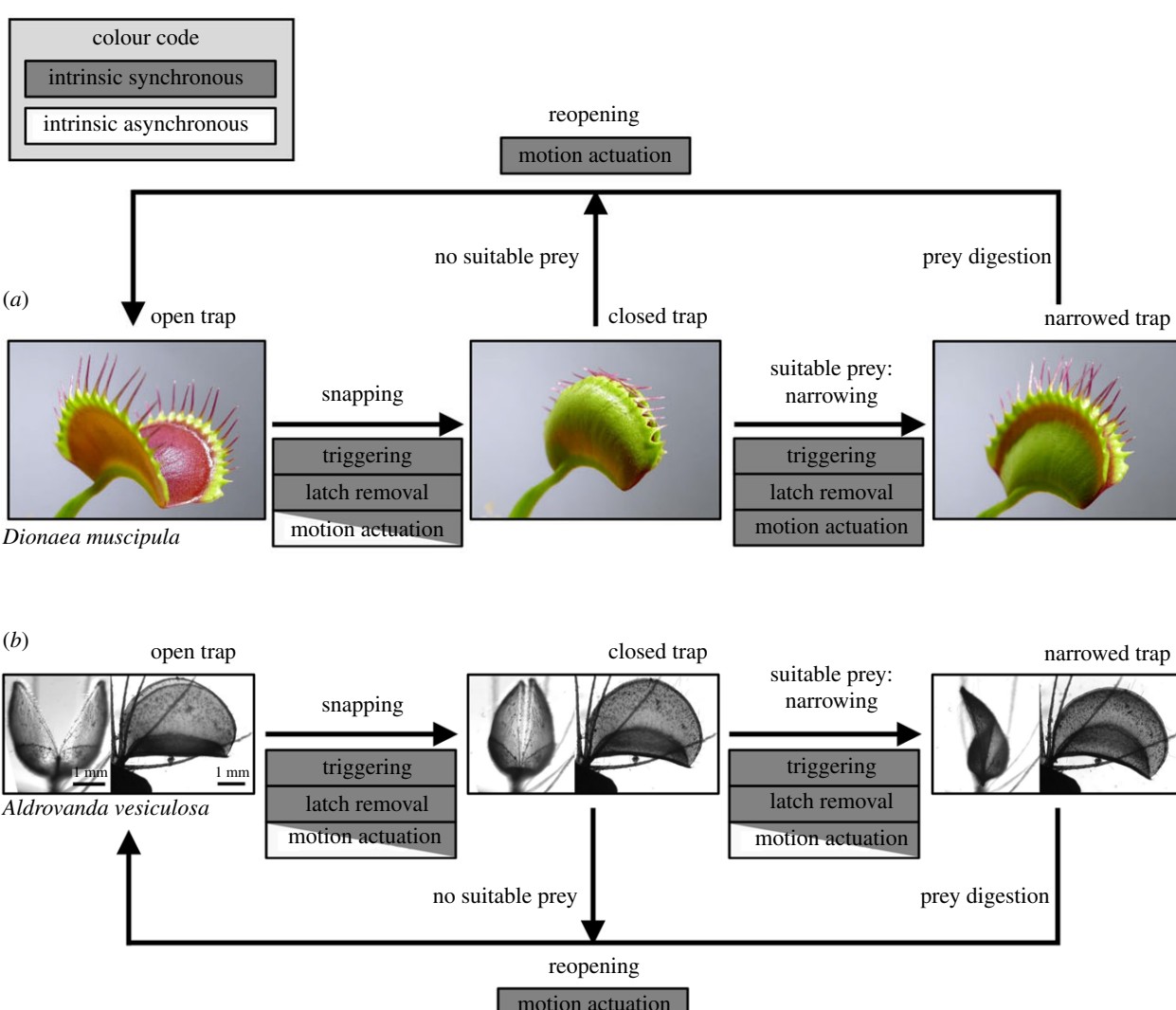

**Figure 1.** Self-actuated snap traps in the Venus flytrap (*D. muscipula*) (*a*) and waterwheel plant (*A. vesiculosa*) (*b*). Both species power all steps of the capture process intrinsically. In *A. vesiculosa*, latch removal is powered synchronously (hydraulic process), but motion actuation during snapping and narrowing combines synchronously and asynchronously powered processes (hydraulics, elastic energy release). In *D. muscipula*, trap narrowing is actuated solely by synchronously powered hydraulic processes; in *Aldrovanda*, the elastic energy release is also at play. Both species reopen their traps slowly by synchronously powered hydraulics. Sub-figure (*a*) modified from [36], in (*b*) from [49]. (Online version in colour.)

trapping cycle: trap setting, triggering and latch removal, and trapping movement.

## (b) Example 2: extrinsic triggering of intrinsically powered motion in asynchronous suction traps

The underwater suction traps of bladderworts (*Utricularia*) (figure 2; electronic supplementary material, table S1) are the fastest of all motile carnivorous plant traps [47,65]. Like snap traps, they typically catch one prey item at a time and can trap repeatedly [66]. Again, the ultra-fast trapping motion results from the sudden release of stored elastic energy [8], but without additional intrinsic energetic costs for triggering. The trap has two kinematic elements, the hollow 'bladder' with elastic lateral walls and the trapdoor that seals the trap entrance. Both elements need to move as a functional unit for successful prey capture [67].

As in snap traps, setting the trap is intrinsically powered. Water is continually pumped out of the trap lumen [68], leading to a sub-ambient hydrostatic pressure of up to 0.14 bar inside the trap [69–71]. The pumping mechanism (i.e. the motor) is not yet fully understood but appears to rely on

respiration-driven, energy-consuming counter-transport of ions and water [69,70,72]. As the sub-ambient pressure inside the trap builds up, the lateral walls (i.e. the springs) bulge inwards, accumulating elastic energy (figure 2*a,e*). Owing to its dome-shaped geometry, the trapdoor resists the sub-ambient pressure and keeps the trap sealed [47]. Trap opening is triggered purely mechanically when prey touch the stiff trigger hairs on the outside of the trapdoor, causing it to deform and buckle inwards [39,47,64]. Water is sucked into the trap lumen at extreme accelerations [8,47,64,73], causing the trap walls to buckle outwards. The trapdoor swings back elastically and closes via reverse snap-buckling, and the trapping cycle starts anew [67]. Bladderwort traps can also fire spontaneously [47,74] via intrinsic actuation (the door buckles under the trap's pressure); however, this extraordinary behaviour is not yet fully understood either mechanistically or in terms of its ecological relevance. So, in contrast to snap traps, where prey triggers an intrinsically powered signalling cascade, the triggering and latch removal in bladderworts is extrinsically actuated by the prey: the trigger hairs act as mechanical levers that locally bend the door and initiate buckling. Hence, the loading of the bladderwort trap is an

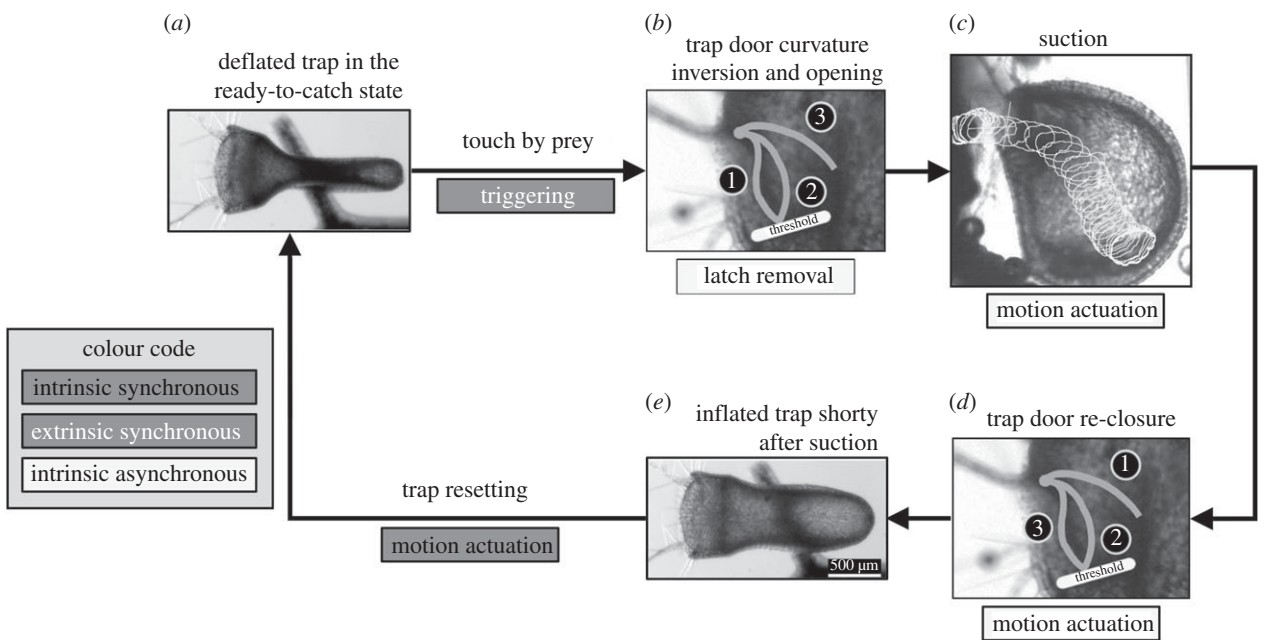

**Figure 2.** The suction traps of bladderworts (*Utricularia* spp.). (*a*) Top view of the set trap. Note the difference in trap volume owing to the deformation of the elastic sidewalls compared with (*e*) immediately after suction. (*b*) Prey bends the trigger hairs on the outer surface of the trapdoor, causing the dome-shaped trap door to snap-buckle and open. The different door configurations are indicated: (i) pre-trigger trapdoor; (ii) inverted curvature after triggering; and (iii) fully opened door. Triggering is powered extrinsically and synchronously by the prey, but latch removal is powered intrinsically and asynchronously by releasing the door's pre-stress during snap-buckling. The process is further accelerated by the inflowing water, driven by the sub-ambient pressure inside the trap. (*c*) Prey (in this case a daphnid, white outline traced at 0.1 ms intervals) is sucked into the trap. Trap door re-closure (*d*) is self-actuated by its intrinsic elastic reset force. (*e*) Immediately after suction, the trap is fully inflated and unable to catch further prey. It is reset though intrinsically powered glands in the trap walls that remove water from the trap interior and pre-stress the trap walls in the process; this resetting takes at least 15 min. After that, the trap is able to catch further prey. (*a*) and (*e*) are modified from [47], (*b*), (*c*) and (*d*) from [64]. (Online version in colour.)

energy-requiring process [74], but the ultra-fast suction occurs without further energy investment and control opportunity for the plant.

## (c) Example 3: extrinsically powered triggering and motion in springboard-pitfall traps

The tropical slender pitcher plant (*N. gracilis*) is currently the only known carnivorous plant that uses an external energy source to actuate a fast trapping motion [46,75] (figure 3; electronic supplementary material, table S1). The kinetic energy of falling raindrops is directly transferred into trap movement. The construction costs of the trap are the only investment by the plant, which are low in comparison to photosynthetically active leaves, and similar to the cost of growing non-motile pitfall traps in other species [76]. A common feature of *Nepenthes* pitcher traps is a canopy-like lid which protects the trap against flooding by rain. In *N. gracilis*, this lid is adapted to work as an impact-powered torsion spring, flicking insects into the pitcher during tropical downpours. This unusual trapping mechanism is based on the combination of three structural adaptations [75]: (i) the lid itself is stiff, but the tissue between pitcher and lid bends easily; (ii) epicuticular wax crystals on the lower lid surface reduce the attachment forces of insects; yet the surface provides sufficient grip for insects in the absence of perturbations; and (iii) the lid is oriented horizontally and positioned closely above the trap opening, maximizing the chance that dislodged insects fall into the pitcher. Together, these adaptations enable *N. gracilis* to exploit the impact force of large raindrops as a power source to drive a fast trapping movement. Neither physiological motors nor

trigger-dependent latches are required, and the movement itself is a completely passive mechanical response to an external impact. In close analogy to the prey capture mechanism of *N. gracilis*, splash-cup plants (e.g. *Chrysosplenium* sp. and *Mazus* sp.) exploit raindrop impacts to disperse their seeds from the bottom of specially adapted cup-shaped flower structures [77], whereas other plants exploit raindrops, wind or passing-by animals to actuate small elastically deformable catapults for diaspore dispersal [78].

## 4. Control over movement comes at the cost of additional energy requirements

Energetic investment in motile trapping mechanisms varies considerably between carnivorous taxa. Why do we not see extrinsically powered trapping motions more widely? The reason could be that relying on extrinsic power means less control over the trapping process. The springboard lid of *N. gracilis* is actuated by a falling raindrop. As such, the trap action is not only out of the plant's control, but also independent of the presence of prey. The high frequency and erratic occurrence of rainstorms in the tropical habitats of *N. gracilis* may increase the odds for successful capture, but the only direct influence that the plant can exert is an increased investment in nectar secretion in order to attract more prey to the underside of the lid [46]. Nevertheless, the plant cannot control if visitors will be present during a drop impact. It should be noted that *N. gracilis*, like all other pitcher plants, possesses additional non-motile trapping mechanisms based on slippery surfaces lining the trap mouth and interior [79,80]. The motion-based lid trapping

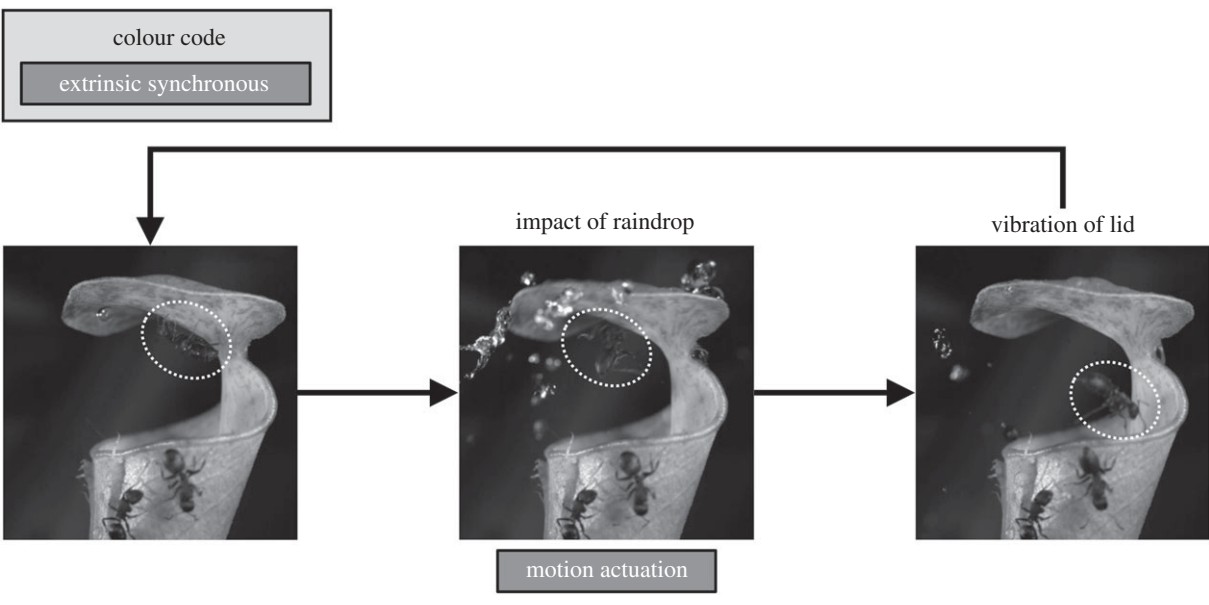

**Figure 3.** The extrinsically actuated motile springboard-pitfall trap of *N. gracilis* does not require trigger structures, motors and latches to function. The movement processes are powered extrinsically (i.e. by the impact of raindrops) and synchronously. An ant foraging on the underside of the pitcher lid (highlighted by a dashed white line) is dislodged by the rapid downward acceleration of the lid and falls into the fluid-filled trap. Figure modified from [36].

mechanism is a supplementary strategy, unlike those of other carnivorous taxa like *Aldrovanda*, *Dionaea* and *Utricularia* that depend exclusively on motile traps.

At the other end of the spectrum, the snapping mechanisms of the Venus flytrap and the waterwheel plant involve multiple intrinsically powered hydraulic processes: (i) latch removal, entailing the release of initially stored pre-stress; (ii) rapid snapping; and (iii) slow final closure of the trap, each providing an opportunity for the plant to fine-tune trap function. Because the setting of the trap, latch removal and initial motion actuation are intrinsically powered, the plant can control all of these processes. This is evident in the famous ability of the Venus flytrap to 'count': at least two instances of trigger hair bending within a restricted time frame are needed to overcome an electrophysiological threshold and initiate the signalling cascade that leads to a snapping event. By integrating multiple successive mechanosensory signals over time, Venus flytraps minimize the risk of 'dryfiring' a trap when there is no prey [52,81–83]. After snapping, continued stimulation is needed to induce the (intrinsically powered) formation of a tightly sealed 'stomach' [43,84]. In the absence of further action potentials, the traps soon reopen.

This high level of control not only helps to avoid false triggering and unnecessary initiation of costly digestion processes after unsuccessful trapping attempts, but also allows the plant to respond to physiological and environmental stresses and demands. Water-stressed Venus flytraps reduce their sensitivity to mechanical stimulation, presumably because the digestion process is strongly water dependent [85]. Environmental factors may also pose constraints on intrinsic movement control. *Aldrovanda* traps show a reduced snapping speed in response to lower water temperature [49,58]. The traps' temperature sensitivity is caused by the hydraulically powered aspects of the trap movement: the elastic response of the midrib is the same in both temperature regimes. This temperature insensitivity of elastically powered processes is consistent with observations in animal predators [14] and indicates a relatively

big influence of hydraulics on the overall motion performance in *Aldrovanda*.

Bladderworts employ extrinsically powered triggering. Thereby, they forgo the adjustability of their trap response, but still ensure that the trap is activated in the presence of prey. Aquatic *Utricularia* are unlikely to experience water stress and might not need to restrict the responsiveness of their traps. However, the traps of non-aquatic bladderworts (e.g. terrestrials, epiphytes) are not constantly surrounded by water and, therefore, may become short-circuited by the aspiration of air bubbles. Several physiological, structural and mechanical features of the trapdoor region have been interpreted as countermeasures to this risk, e.g. appendages that might hold a water reservoir in front of the trap mouth [86,87].

## 5. How costly are physiological processes?

The universal limitation of biological processes by available energy, in the 'currency' of ATP, is a major evolutionary driver and has shaped organisms, including their sensory and motor control systems [88]. This limitation is somewhat relaxed in green plants that can photosynthesize carbohydrates to 'fuel' their physiological processes. Carnivorous plants predominantly colonize sun-exposed, permanently wet habitats such as bogs, tropical cloud forests and aquatic habitats, where energy conservation might not be under strong selective pressure. Indeed, carnivorous plants have been shown to be nutrient- rather than energy-limited [89]. In order to understand the costs and benefits of different motile mechanisms, we need to quantify and compare energy consumption and nutrient uptake rates. Does the investment of carbohydrates in attractive nectar in *N. gracilis* match the costs of electrophysiological signalling in *D. muscipula* or the constant running of the 'water pump' in *Utricularia*? What is the pay-off of different trapping mechanisms? The prey-triggered action of snap traps and suction bladders increases the success rate of the individual 'strike' but comes at the cost of an inactive resetting phase. By contrast,

the opportunistic *N. gracilis* pitcher can trap continually, as long as rainfall persists [75]. Snap traps typically catch one prey item at a time, but can capture exceptionally large individuals [51,90]. The traps remain tightly sealed during prey digestion and can therefore not trap further prey during this period [58,91]. The bladders of *Utricularia* typically also trap individual prey, but in contrast with snap traps, they reset quickly after prey capture, and further prey can be caught while digestion is ongoing [66]. The springboard trap of *N. gracilis* can not only catch multiple insects in a single strike, but the slippery surfaces of the pitfall trap may trap further prey at the same time. As in all pitfall traps, prey accumulates and is digested continually in the fluid pool at the bottom of the pitcher.

## 6. Complexity and diversity of trapping systems

The examples above highlight the varying levels of energy investment in movement as well as the diversity of realized actuation mechanisms. When looking at the entirety of carnivorous plant traps (electronic supplementary material, table S1), an additional level of complexity becomes apparent because movement is regularly employed alongside other, non-motile mechanisms such as pitfall, glue or eel trapping [39]. We already saw that *N. gracilis* combines the movement-based, but extrinsically actuated lid trapping mechanism [46] with the use of non-motile, slippery pitfall traps akin to those of other *Nepenthes* species, Sarraceniaceae, *Cephalotus follicularis* and carnivorous Bromeliads. Several species of sundews, e.g. *D. glanduligera*, employ fast-moving 'catapulting' snap tentacles in combination with sticky 'flypaper' secretions to capture prey [38]. In *D. glanduligera*, fast tentacle movement is irreversible while the (slower) tentacles of several other species are capable of resetting and snapping repeatedly [36]. Moreover, movement is employed not only for the act of trapping itself, but may also help to secure and digest already captured prey, as in the slow-moving 'flypaper' leaves of motile sundews (*Drosera*) and butterworts (*Pinguicula*) that engulf captured insects in digestive mucus, or it may kill the prey, as in the crushing 'stomachs' formed by *Dionaea* and *Aldrovanda* [36,39,91]. Hence, different species use combinations of motile and non-motile structures as well as intrinsically or extrinsically powered processes to catch, retain and kill their prey.

The astonishing diversity of trap functions and mechanisms makes carnivorous plants ideal candidates for the study of plant movement. Especially traps with 'outlier' functions, i.e. those that differ from the *status quo* of their congeners, can provide valuable insights into the underlying mechanisms of movement. Comparison of lid kinematics, surface texture, and experimental trapping success between *N. gracilis* and a closely related non-motile pitfall trapping species, *Nepenthes rafflesiana*, revealed the key adaptations underlying the lid spring mechanism [75]. By comparing snap tentacles with other, slow-moving *Drosera* tentacles,

we may unravel the morphological and physiological adaptations that enable fast tentacle movement [38,92]. Equally, comparing resettable versus 'single use' snap tentacles may elucidate trade-offs between movement speed and durability of trapping structures. Within the bladderworts, *Utricularia multifida* might be such an insight-providing outlier, because its traps did not show any firing events, or any water pumping activity in laboratory studies [86,93]. Hence, it is speculated that this species possesses non-motile eel traps similar to closely related corkscrew plants (*Genlisea*) [94].

## 7. Conclusion and outlook

Using three examples, we highlighted the diversity of motion amplification and control strategies in motile carnivorous plant traps. We discussed the multitier investment of metabolic energy to power individual capture events in the context of frameworks for animal movement. The traditional distinction of 'active' versus 'passive' carnivorous plant traps (*sensu* [39]) obscured this functional diversity and failed to capture the underlying trade-off between structural and metabolic investment on the one hand, and control over the trapping process on the other [95,96] (electronic supplementary material, table S1).

The similarities between fast animal and plant motions are striking: both organismal groups employ 'mechanical tricks' in the form of springs to overcome innate limitations of muscles or hydraulics. Although not discussed here, the same holds true for fast fungal movement, e.g. in the context of spore discharge [97,98]. Comparative cross-kingdom analyses of the functional principles and resilience of the various mechanisms as well as their evolutionary pathways and ecological significances constitute important aspects for future studies.

Data accessibility. This article has no additional data.

Authors' contributions. All authors gave final approval for publication and agreed to be held accountable for the work performed therein.

Competing interests. The authors declare no competing interests.

Funding. U.B. is funded by a Royal Society University Research Fellowship (UF150138) and research grants from the Royal Society (RGF/EA/180059 and RGF/R1/180034) and the Human Frontiers Science Programme (RGY0082/2021). U.K.M. acknowledges funding by the National Science Foundation (award number NSF-BIO-IOS no. 1352130). S.P. receives funding from the academic research alliance JONAS (Joint Research Network on Advanced Materials and Systems). S.P. further acknowledges funding by the Deutsche Forschungsgemeinschaft (DFG, German Research Foundation) under Germany's Excellence Strategy – EXC-2193/1 – 390951807.

Acknowledgements. This paper was inspired by fruitful discussions between the authors following a New Phytologist Trust-funded workshop at Harvard Forest, MA, in October 2018. U.K.M. and U.B. thank Jon Millet and Aaron Ellison for organizing and inviting them to this workshop. Stephen Deban helped identify suitable animal examples. Furthermore, all authors gratefully acknowledge the constructive feedback by three anonymous referees on an earlier version of this article.

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
