## [Peer Review File · Proceedings of the Royal Society B: Biological Sciences]

Review History

RSPB-2020-0515.R0 (Original submission)

Review form: Reviewer 1

Recommendation

Accept with minor revision (please list in comments)

Scientific importance: Is the manuscript an original and important contribution to its field?

Acceptable

General interest: Is the paper of sufficient general interest?

Acceptable

Quality of the paper: Is the overall quality of the paper suitable?

Acceptable

Is the length of the paper justified?

Yes

Should the paper be seen by a specialist statistical reviewer?

No

Do you have any concerns about statistical analyses in this paper? If so, please specify them explicitly in your report.

No

It is a condition of publication that authors make their supporting data, code and materials available - either as supplementary material or hosted in an external repository. Please rate, if applicable, the supporting data on the following criteria.

Is it accessible?

N/A

Is it clear?

N/A

Is it adequate?

N/A

Do you have any ethical concerns with this paper?

No

Comments to the Author

Carnivorous plants have been associated inherently with rapid movements of their traps for long time. Besides this, there are also commonly slow movements of traps of carnivorous plants. The literature has usually termed the traps “active” and “passive” as based on the expenditure of metabolic energy to substantiate/drive the (rapid) movements, although all traps do function with the expenditure of metabolic energy (and also organic and mineral nutrients) not only for casual movement, but also for the trap construction, maintenance, enzyme secretion and nutrient absorption. As a result of this, the terms “active” and “passive” lose their sense both generally and also in the context of trap movements. The team of experienced authors, working significantly and successfully in the field of mobile traps, has tried to critically review and discuss the topics of activity and mobility of traps and has suggested to abandon the previously accepted subdivision of carnivorous plant traps into active and passive. I want to add that all primary literature data concerning CP traps have been published (at least a few years ago) and are not quite novel. Yet I consider the present paper as original as it thoroughly explains the interrelationships between trap motility and activity. In this respect, this well-arranged review is very useful and beneficial for the readers not only dealt with carnivorous plants or rapid plant movements. The three typical examples of rapid trap movements selected document the dominant majority of rapid movements in CPs. I have found only a few minor mistakes or typos to be corrected. They are listed below (page/line):

p.5, l.95: one prey animal?: maybe better: one animal prey;

p.5, l.117: “The traps of both species can snap repeatedly and re-open actively by growth” . Is this really so for *Aldrovanda*, too? I have thought that the re-opening in this species is a turgor-based process. See also p.27, l. 588.

p.10, l.231-232: Carnivorous plants exclusively colonize sun-exposed, permanently wet habitats.... : I suppose that the word “exclusively” is not true as there are several dozens of CP species growing typically in shade. I suggest to write instead: mainly or usually or dominantly or prevailingly. Within this paragraph, the authors could also mention that terrestrial CPs grow relatively slowly, which reduces their needs for energy [17].

p.12, l.274: status quo: perhaps in italics?

p.15, l.360: should be: 17. Ellison AM,;

p. 22, 1,505: 69. Niven JE & McLaughlin SB. 2008: should it be with or without „&”?

References: some journal names are abbreviated, some are not. Is it OK? To be united?

The photos properly document the text of the paper. However, Table 1S has not been available to me.

In conclusion, the paper is an interesting contribution to the knowledge of carnivorous plant biophysics and physiology, but addresses a broader readership and should be published after a minor revision, mostly formal.

Review form: Reviewer 2

Recommendation

Reject – article is not of sufficient interest (we will consider a transfer to another journal)

Scientific importance: Is the manuscript an original and important contribution to its field?

Good

General interest: Is the paper of sufficient general interest?

Acceptable

Quality of the paper: Is the overall quality of the paper suitable?

Good

Is the length of the paper justified?

Yes

Should the paper be seen by a specialist statistical reviewer?

No

Do you have any concerns about statistical analyses in this paper? If so, please specify them explicitly in your report.

No

It is a condition of publication that authors make their supporting data, code and materials available - either as supplementary material or hosted in an external repository. Please rate, if applicable, the supporting data on the following criteria.

Is it accessible?

N/A

Is it clear?

N/A

Is it adequate?

N/A

Do you have any ethical concerns with this paper?

No

Comments to the Author

This manuscript was enjoyable to read and presents some proposed terminological changes to how trapping types are defined in carnivorous plants. It is well-written and with few grammatical or other errors ('divers' on L30 was the only obvious error). The rationale for changing terminology is sound and sufficiently justified. However, I am not sure that the content, potential authorship, or novelty of the theory presented meet the journal's requirement for reviews to be of "outstanding scientific importance and broad general interest". Certainly plant physiologists study carnivorous plants will find the review of interest, and the proposed terminology will likely be adopted within the carnivorous plant literature, but I am not sure I can see a significantly broader potential readership (e.g., population ecology, evolutionary ecology, botany) than this. I note the author's comments in the introduction that consistent terminology is important, and I certainly see the value in what is being proposed, but I feel it may be more suited to a journal publishing shorter reviews of this nature such as Biology Letters.

Review form: Reviewer 3

Recommendation

Accept with minor revision (please list in comments)

Scientific importance: Is the manuscript an original and important contribution to its field?

Good

General interest: Is the paper of sufficient general interest?

Good

Quality of the paper: Is the overall quality of the paper suitable?

Excellent

Is the length of the paper justified?

Yes

Should the paper be seen by a specialist statistical reviewer?

No

Do you have any concerns about statistical analyses in this paper? If so, please specify them explicitly in your report.

No

It is a condition of publication that authors make their supporting data, code and materials available - either as supplementary material or hosted in an external repository. Please rate, if applicable, the supporting data on the following criteria.

Is it accessible?

Yes

Is it clear?

Yes

Is it adequate?

Yes

Do you have any ethical concerns with this paper?

No

Comments to the Author

This is a generally well-written, interesting, and informative manuscript concerning the diversity of motile carnivorous plant traps. The authors use three diverse but well-studied examples to argue that the energetic cost of movement and resetting of traps varies dramatically, and therefore that use of the terms "motile" and "nonmotile" would be more accurate than "active" and "passive."

Although the major thesis of the manuscript is clear, the argument is sometimes confusing. This is because the authors themselves employ the terms "active" and "passive" repeatedly in describing the movement in the three examples that they use (see, for example, lines 78-80). It seems that they use these terms in reference to energy expenditure (lines 63-66). However, it would be helpful if they explicitly defined their use of the terms "active" and "passive" as "energy-requiring" or "energetically cost free." Even better, in my opinion, would be to avoid use of the terms "active" and "passive" completely while making the argument that these terms are misleading. They could instead use "energy-consuming" or "non-energy-consuming."

Because of these issues, it seems that the authors aren't really arguing against the use of the terms "active" and "passive," but instead the recognition of the distinction between "active" and "motile".

There are a few minor typographical and/or editorial errors that need attention:

Line 30: I assume that "divers" should be "diverse."

Lines 73-74: The clause at the end of this sentence ("yet vary considerably in their investment in active processes proceeding such motion") doesn't make sense. Should "proceeding" be changed to "preceding"?

Decision letter (RSPB-2020-0515.R0)

04-Jun-2020

Dear Mr Poppinga:

I am writing to inform you that your manuscript RSPB-2020-0515 entitled "Functional and mechanical diversity of motile carnivorous plants, or why we should stop talking of 'active' versus 'passive' traps" has, in its current form, been rejected for publication in Proceedings B.

This action has been taken on the advice of referees, who have recommended that substantial revisions are necessary. With this in mind we would be happy to consider a resubmission, provided the comments of the referees are fully addressed. However please note that this is not a provisional acceptance.

A concern is that the review does not have broad enough appeal for Proc B's general readership. Terminological changes can be truly helpful, especially to specialists, but the undertone of the reviews is that something more is needed for this review to excite a broad audience. It is important that one or more reviewers is won over if you choose to resubmit, or we may have to make the undesirable decision of rejection. Nonetheless this is an intriguing topic and we hope that it can fit Proc B.

Sincerely,
John R Hutchinson, Editor
mailto: proceedingsb@royalsociety.org

Associate Editor
Reviewer(s)' Comments to Author:

Referee: 1

Comments to the Author(s)

Carnivorous plants have been associated inherently with rapid movements of their traps for long time. Besides this, there are also commonly slow movements of traps of carnivorous plants. The literature has usually termed the traps "active" and "passive" as based on the expenditure of metabolic energy to substantiate/drive the (rapid) movements, although all traps do function with the expenditure of metabolic energy (and also organic and mineral nutrients) not only for casual movement, but also for the trap construction, maintenance, enzyme secretion and nutrient absorption. As a result of this, the terms "active" and "passive" lose their sense both generally and also in the context of trap movements. The team of experienced authors, working significantly and successfully in the field of mobile traps, has tried to critically review and discuss the topics of activity and mobility of traps and has suggested to abandon the previously accepted subdivision of carnivorous plant traps into active and passive. I want to add that all primary literature data concerning CP traps have been published (at least a few years ago) and are not quite novel. Yet I consider the present paper as original as it thoroughly explains the interrelationships between trap motility and activity. In this respect, this well-arranged review is very useful and beneficial for the readers not only dealt with carnivorous plants or rapid plant movements. The three typical examples of rapid trap movements selected document the dominant majority of rapid movements in CPs. I have found only a few minor mistakes or typos to be corrected. They are listed below (page/line):

p.5, l.95: one prey animal?: maybe better: one animal prey;

p.5, l.117: "The traps of both species can snap repeatedly and re-open actively by growth" . Is this really so for *Aldrovanda*, too? I have thought that the re-opening in this species is a turgor-based process. See also p.27, l. 588.

p.10, l.231-232: Carnivorous plants exclusively colonize sun-exposed, permanently wet habitats.... : I suppose that the word "exclusively" is not true as there are several dozens of CP species growing typically in shade. I suggest to write instead: mainly or usually or dominantly or

prevailingly. Within this paragraph, the authors could also mention that terrestrial CPs grow relatively slowly, which reduces their needs for energy [17].

p.12, l.274: status quo: perhaps in italics?

p.15, l.360: should be: 17. Ellison AM,;

p. 22, l.505: 69. Niven JE & McLaughlin SB. 2008: should it be with or without „&“?.

References: some journal names are abbreviated, some are not. Is it OK? To be united?

The photos properly document the text of the paper. However, Table 1S has not been available to me.

In conclusion, the paper is an interesting contribution to the knowledge of carnivorous plant biophysics and physiology, but addresses a broader readership and should be published after a minor revision, mostly formal.

Referee: 2

Comments to the Author(s)

This manuscript was enjoyable to read and presents some proposed terminological changes to how trapping types are defined in carnivorous plants. It is well-written and with few grammatical or other errors ('divers' on L30 was the only obvious error). The rationale for changing terminology is sound and sufficiently justified. However, I am not sure that the content, potential authorship, or novelty of the theory presented meet the journal's requirement for reviews to be of "outstanding scientific importance and broad general interest". Certainly plant physiologists study carnivorous plants will find the review of interest, and the proposed terminology will likely be adopted within the carnivorous plant literature, but I am not sure I can see a significantly broader potential readership (e.g., population ecology, evolutionary ecology, botany) than this. I note the author's comments in the introduction that consistent terminology is important, and I certainly see the value in what is being proposed, but I feel it may be more suited to a journal publishing shorter reviews of this nature such as *Biology Letters*.

Referee: 3

Comments to the Author(s)

This is a generally well-written, interesting, and informative manuscript concerning the diversity of motile carnivorous plant traps. The authors use three diverse but well-studied examples to argue that the energetic cost of movement and resetting of traps varies dramatically, and therefore that use of the terms "motile" and "nonmotile" would be more accurate than "active" and "passive."

Although the major thesis of the manuscript is clear, the argument is sometimes confusing. This is because the authors themselves employ the terms "active" and "passive" repeatedly in describing the movement in the three examples that they use (see, for example, lines 78-80). It seems that they use these terms in reference to energy expenditure (lines 63-66). However, it would be helpful if they explicitly defined their use of the terms "active" and "passive" as "energy-requiring" or "energetically cost free." Even better, in my opinion, would be to avoid use of the terms "active" and "passive" completely while making the argument that these terms are misleading. They could instead use "energy-consuming" or "non-energy-consuming."

Because of these issues, it seems that the authors aren't really arguing against the use of the terms "active" and "passive," but instead the recognition of the distinction between "active" and "motile".

There are a few minor typographical and/or editorial errors that need attention:

Line 30: I assume that "divers" should be "diverse."

Lines 73-74: The clause at the end of this sentence ("yet vary considerably in their investment in active processes proceeding such motion") doesn't make sense. Should "proceeding" be changed to "preceding"?

Author's Response to Decision Letter for (RSPB-2020-0515.R0)

See Appendix A.

RSPB-2021-0771.R0

Review form: Reviewer 1 (Lubomír Adamec)

Recommendation

Accept with minor revision (please list in comments)

Scientific importance: Is the manuscript an original and important contribution to its field?

Excellent

General interest: Is the paper of sufficient general interest?

Acceptable

Quality of the paper: Is the overall quality of the paper suitable?

Excellent

Is the length of the paper justified?

Yes

Should the paper be seen by a specialist statistical reviewer?

No

Do you have any concerns about statistical analyses in this paper? If so, please specify them explicitly in your report.

No

It is a condition of publication that authors make their supporting data, code and materials available - either as supplementary material or hosted in an external repository. Please rate, if applicable, the supporting data on the following criteria.

Is it accessible?

N/A

Is it clear?

N/A

Is it adequate?

N/A

Do you have any ethical concerns with this paper?

No

Comments to the Author

In this review paper, experienced authors deal with rapid movements of traps of carnivorous plants from a biophysical and functional point of view. They are focused on snap traps, suction traps and springboard-pitfall traps all exhibiting remarkable rapid movements. They distinguish three aspects of what has previously been named in the terms 'active' and 'passive' movement: (1) motion, (2) the conversion of metabolic energy into movement, (3) and the relative timing of energy conversion and motion. Although much has been written on rapid movements of traps of carnivorous plants, such an approach seems to be novel and promising for future research in this field. I have found that this well-arranged paper is perfectly written and reads well, underlining substantial features and aspects and omitting details. Moreover, the English is perfect and well understandable. Thus, I have found only a few minor mistakes/typos to be corrected or commented (page/line):

p.7., l.130: "by"? should be probably: "be";

p.11, l. 226-241: Here, the authors have forgotten to state one crucial trait of Utricularia traps: spontaneous firing without any extrinsic stimulus, only when the negative pressure inside the bladder reaches the "critical" pressure value and the trapdoor cannot resist this pressure anymore. It is evident that this spontaneous firing is i) universal in aquatic Utricularia species and ii) has some ecological importance to gain suspended particles (e.g., nanoplankton) from the ambient water instead of prey. However, in this case, the triggering is entirely intrinsically powered. Thus, the authors should include this aspect of these traps into the text here.

p.15, l.319: "physiology"? *Sensu stricto*, physiology is a scientific branch. The authors might want to say rather: "physiological processes".

p.15, l.324: "nutrient intake"? perhaps better and more standardly: "uptake";

p.23, l.492: add volume and pages;

p.23, l.497: Development (capital "D");

p.24, l.519: dynamicks? Correctly: dynamics;

p.24, l.540: Correctly: PloS One;

p.25, l.554: Nature Plants;

p.25, l.559, p.27, l.612, p.28, l.623, p.28, l.637: correctly: of Sciences of the USA;

p.27, l.605: Scientific Reports;

p.28, l.620: the second parenthesis?

p.28, l.628: volume??

p.30, 31: Legends to Figs. 1 and 2: species names in italics.

In conclusion, the paper needs only a very minor revision.

Decision letter (RSPB-2021-0771.R0)

26-Apr-2021

Dear Mr Poppinga

I am pleased to inform you that your manuscript RSPB-2021-0771 entitled "Complexity and diversity of motion amplification and control strategies in motile carnivorous plant traps" has been accepted for publication in Proceedings B. Congratulations!!

The referee(s) have recommended publication, but also suggest some minor revisions to your manuscript. Therefore, I invite you to respond to the referee(s)' comments and revise your manuscript. Because the schedule for publication is very tight, it is a condition of publication that you submit the revised version of your manuscript within 7 days. If you do not think you will be able to meet this date please let us know.

The reviewer has made some helpful final suggestions regarding wording and other presentational issues that all seem easy to implement.

[http://datadryad.org/submit?journalID=RSPB&manu=\(Document not available\)](http://datadryad.org/submit?journalID=RSPB&manu=(Document not available)) which will take you to your unique entry in the Dryad repository. If you have already submitted your data to dryad you can make any necessary revisions to your dataset by following the above link.

Please see <https://royalsociety.org/journals/ethics-policies/data-sharing-mining/> for more details.

Sincerely,

Prof. John R. Hutchinson, Editor

Reviewer(s)' Comments to Author:

Referee: 1

Comments to the Author(s)

In this review paper, experienced authors deal with rapid movements of traps of carnivorous plants from a biohysical and functional point of view. They are focused on snap traps, suction traps and springboard-pitfall traps all exhibiting remarkable rapid movements. They distinguish three aspects of what has previously been named in the terms 'active' and 'passive' movement: (1) motion, (2) the conversion of metabolic energy into movement, (3) and the relative timing of energy conversion and motion. Although much has been written on rapid movements of traps of carnivorous plants, such an approach seems to be novel and promising for future research in this field. I have found that this well-arranged paper is perfectly written and reads well, underlining substantial features and aspects and omitting details. Moreover, the English is perfect and well understandable. Thus, I have found only a few minor mistakes/typos to be corrected or commented (page/line):

p.7., l.130: "by"? should be probably: "be";

p.11, l. 226-241: Here, the authors have forgotten to state one crucial trait of Utricularia traps: spontaneous firing without any extrinsic stimulus, only when the negative pressure inside the bladder reaches the "critical" pressure value and the trapdoor cannot resist this pressure anymore. It is evident that this spontaneous firing is i) universal in aquatic Utricularia species and ii) has some ecological importance to gain suspended particles (e.g., nanoplankton) from the ambient water instead of prey. However, in this case, the triggering is entirely intrinsically powered. Thus, the authors should include this aspect of these traps into the text here.

p.15, l.319: "physiology"? *Sensu stricto*, physiology is a scientific branch. The authors might want to say rather: "physiological processes".

p.15, l.324: "nutrient intake"? perhaps better and more standardly: "uptake";

p.23, l.492: add volume and pages;

p.23, l.497: Development (capital "D");

p.24, l.519: dynamicks? Correctly: dynamics;

p.24, l.540: Correctly: PloS One;

p.25, l.554: Nature Plants;

p.25, l.559, p.27, l.612, p.28, l.623, p.28, l.637: correctly: of Sciences of the USA;

p.27, l.605: Scientific Reports;

p.28, l.620: the second parenthesis?

p.28, l.628: volume??

p.30, 31: Legends to Figs. 1 and 2: species names in italics.

In conclusion, the paper needs only a very minor revision.

Author's Response to Decision Letter for (RSPB-2021-0771.R0)

See Appendix B.

Decision letter (RSPB-2021-0771.R1)

30-Apr-2021

Dear Mr Poppinga

I am pleased to inform you that your manuscript entitled "Complexity and diversity of motion amplification and control strategies in motile carnivorous plant traps" has been accepted for publication in Proceedings B.

If you are likely to be away from e-mail contact during this period, let us know. Due to rapid publication and an extremely tight schedule, if comments are not received, we may publish the paper as it stands.

Data Accessibility section

Open access

You are invited to opt for open access via our author pays publishing model. Payment of open access fees will enable your article to be made freely available via the Royal Society website as soon as it is ready for publication. For more information about open access publishing please visit our website at http://royalsocietypublishing.org/site/authors/open_access.xhtml.

The open access fee is £1,700 per article (plus VAT for authors within the EU). If you wish to opt for open access then please let us know as soon as possible.

Paper charges

Sincerely,

Proceedings B

Appendix A

Reply to editor and reviewers

Manuscript ID RSPB-2020-0515

Associate Editor:

(1) A concern is that the review does not have broad enough appeal for Proc B's general readership. Terminological changes can be truly helpful, especially to specialists, but the undertone of the reviews is that something more is needed for this review to excite a broad audience. It is important that one or more reviewers is won over if you choose to resubmit, or we may have to make the undesirable decision of rejection. Nonetheless this is an intriguing topic and we hope that it can fit Proc B.

Reply: We agree with the editor's concern. To make the manuscript relevant to a broad readership, we expanded the scope of the manuscript substantially by changing the main narrative and conclusions of the manuscript. The manuscript places carnivorous plants into the framework of latch-mediated springs to explain the complexities and wide range of designs of their traps. So the change in terminology is now a recommendation coming forth from this framework. In the process of implementing this new narrative, we rewrote substantial portions of the manuscript.

Referee: 1

Carnivorous plants have been associated inherently with rapid movements of their traps for long time. Besides this, there are also commonly slow movements of traps of carnivorous plants. The literature has usually termed the traps "active" and "passive" as based on the expenditure of metabolic energy to substantiate/drive the (rapid) movements, although all traps do function with the expenditure of metabolic energy (and also organic and mineral nutrients) not only for casual movement, but also for the trap construction, maintenance, enzyme secretion and nutrient absorption. As a result of this, the terms "active" and "passive" lose their sense both generally and also in the context of trap movements. The team of experienced authors, working significantly and successfully in the field of mobile traps, has tried to critically review and discuss the topics of activity and mobility of traps and has suggested to abandon the previously accepted subdivision of carnivorous plant traps into active and passive. I want to add that all primary literature data concerning CP traps have been published (at least a few years ago) and are not quite novel. Yet I consider the present paper as original as it thoroughly explains the interrelationships between trap motility and activity. In this respect, this well-arranged review is very useful and beneficial for the readers not only dealt with carnivorous plants or rapid plant movements. The three typical examples of rapid trap movements selected document the dominant majority of rapid movements in CPs. I have found only a few minor mistakes or typos to be corrected. They are listed below (page/line):

p.5, l.95: one prey animal?: maybe better: one animal prey.

Reply: The relevant sentence and section have been rewritten entirely during the revision.

p.5, l.117: "The traps of both species can snap repeatedly and re-open actively by growth" . Is this really so for *Aldrovanda*, too? I have thought that the re-opening in this species is a turgor-based process. See also p.27, l. 588.

Reply: The revised manuscript states "The traps of both species can snap repeatedly, and re-open by means of energy-requiring growth processes [57, 63]", which is in fact the same statement. It is true that *Aldrovanda* also re-open by growth (reference 57: Ashida et al., 1934).

p.10, l.231-232: Carnivorous plants exclusively colonize sun-exposed, permanently wet habitats.... : I suppose that the word “exclusively” is not true as there are several dozens of CP species growing typically in shade. I suggest to write instead: mainly or usually or dominantly or prevailingly. Within this paragraph, the authors could also mention that terrestrial CPs grow relatively slowly, which reduces their needs for energy [17].

Reply: The revised manuscript states “Carnivorous plants predominantly colonize sun-exposed, permanently wet habitats such as bogs, tropical cloud forests, and aquatic habitats, where energy conservation might not be under strong selective pressure. Indeed, carnivorous plants have been shown to be nutrient- rather than energy-limited [89].”

p.12, l.274: status quo: perhaps in italics?

Reply: complied

p.15, l.360: should be: 17. Ellison AM,;

Reply: corrected.

p. 22, l.505: 69. Niven JE & McLaughlin SB. 2008: should it be with or without „&”?

Reply: corrected.

References: some journal names are abbreviated, some are not. Is it OK? To be united?

Reply: References have been updated to ensure a uniform style complying with Proc B requirements.

The photos properly document the text of the paper. However, Table 1S has not been available to me.

Reply: We hope that the revised manuscript has all supplementary files available to the reviewer.

In conclusion, the paper is an interesting contribution to the knowledge of carnivorous plant biophysics and physiology, but addresses a broader readership and should be published after a minor revision, mostly formal.

Referee: 2

This manuscript was enjoyable to read and presents some proposed terminological changes to how trapping types are defined in carnivorous plants. It is well-written and with few grammatical or other errors ('divers' on L30 was the only obvious error). The rationale for changing terminology is sound and sufficiently justified. However, I am not sure that the content, potential authorship, or novelty of the theory presented meet the journal's requirement for reviews to be of "outstanding scientific importance and broad general interest". Certainly plant physiologists study carnivorous plants will find the review of interest, and the proposed terminology will likely be adopted within the carnivorous plant literature, but I am not sure I can see a significantly broader potential readership (e.g., population ecology, evolutionary ecology, botany) than this. I note the author's comments in the introduction that consistent terminology is important, and I certainly see the value in what is being proposed, but I feel it may be more suited to a journal publishing shorter reviews of this nature such as Biology Letters.

Reply: We agree with the reviewer's concern, reiterated by the editor. To make the manuscript relevant to a broad readership, we expanded the scope of the manuscript substantially by changing the main narrative and conclusions of the manuscript. The manuscript places carnivorous plants into the framework of latch-mediated springs to explain the complexities and wide range of designs of their

traps. So the change in terminology is now a recommendation coming forth from this framework. In the process of implementing this new narrative, we rewrote substantial portions of the manuscript.

Referee: 3

This is a generally well-written, interesting, and informative manuscript concerning the diversity of motile carnivorous plant traps. The authors use three diverse but well-studied examples to argue that the energetic cost of movement and resetting of traps varies dramatically, and therefore that use of the terms “motile” and “nonmotile” would be more accurate than “active” and “passive.”

Although the major thesis of the manuscript is clear, the argument is sometimes confusing. This is because the authors themselves employ the terms “active” and “passive” repeatedly in describing the movement in the three examples that they use (see, for example, lines 78-80). It seems that they use these terms in reference to energy expenditure (lines 63-66). However, it would be helpful if they explicitly defined their use of the terms “active” and “passive” as “energy-requiring” or “energetically cost free.” Even better, in my opinion, would be to avoid use of the terms “active” and “passive” completely while making the argument that these terms are misleading. They could instead use “energy-consuming” or “non-energy-consuming.”

Because of these issues, it seems that the authors aren’t really arguing against the use of the terms “active” and “passive,” but instead the recognition of the distinction between “active” and “motile”.

Reply: We agree with the reviewer and have revised our terminology throughout. We now void the terms ‘active’ and ‘passive’ entirely and introduce the terms ‘motile’ versus ‘non-motile’, ‘intrinsically powered’ versus ‘extrinsically powered’. We also introduce the terms ‘synchronous’ and ‘asynchronous’ to describe movements that are powered directly by an actuator that converts metabolic energy into kinetic energy, versus movements that are powered by a spring that was loaded by a metabolically powered actuator. We hope that the reviewer will agree with our decision against the term energy-consuming because we want our terminology to align with the thermodynamic fact that energy is strictly speaking converted rather than consumed.

There are a few minor typographical and/or editorial errors that need attention:

Line 30: I assume that “divers” should be “diverse.”

Reply: The text has been revised, eliminating the phrase in question.

Lines 73-74: The clause at the end of this sentence (“yet vary considerably in their investment in active processes proceeding such motion”) doesn’t make sense. Should “proceeding” be changed to “preceding”?

Reply: The text has been revised, eliminating the phrase in question.

Appendix B

Response to referee

In this review paper, experienced authors deal with rapid movements of traps of carnivorous plants from a biohysical and functional point of view. They are focused on snap traps, suction traps and springboard-pitfall traps all exhibiting remarkable rapid movements. They distinguish three aspects of what has previously been named in the terms ‘active’ and ‘passive’ movement: (1) motion, (2) the conversion of metabolic energy into movement, (3) and the relative timing of energy conversion and motion. Although much has been written on rapid movements of traps of carnivorous plants, such an approach seems to be novel and promising for future research in this field. I have found that this well-arranged paper is perfectly written and reads well, underlining substantial features and aspects and omitting details. Moreover, the English is perfect and well understandable. Thus, I have found only a few minor mistakes/typos to be corrected or commented (page/line):

Response: We thank the reviewer very much for their positive and constructive feedback.

p.7., l.130: “by”? should be probably: “be”;

Response: Corrected.

p.11, l. 226-241: Here, the authors have forgotten to state one crucial trait of Utricularia traps: spontaneous firing without any extrinsic stimulus, only when the negative pressure inside the bladder reaches the “critical” pressure value and the trapdoor cannot resist this pressure anymore. It is evident that this spontaneous firing is i) universal in aquatic Utricularia species and ii) has some ecological importance to gain suspended particles (e.g., nanoplankton) from the ambient water instead of prey. However, in this case, the triggering is entirely intrinsically powered. Thus, the authors should include this aspect of these traps into the text here.

Response: We discussed this point among all co-authors and at the insistence of co-author Müller decided initially to not address spontaneous fires for the sake of remaining within the scope of this review by not rising a complex, unresolved issue that we initially judged to be not central to our argument. Clearly, the reviewer disagrees, so our revised text briefly notes the occurrence of spontaneous firings and highlights our still limited understanding of this phenomenon. The revised text now reads:

“Bladderwort traps can also fire spontaneously [47, 74] via intrinsic actuation (the door buckles under the trap’s pressure), however, this extraordinary behaviour is not yet fully understood either mechanistically or in terms of its ecological relevance. So, in contrast to snap traps, where prey trigger an intrinsically powered signalling cascade, the triggering and latch removal in bladderworts is extrinsically actuated by the prey: the trigger hairs act as mechanical levers that locally bend the door and initiate buckling.”

For the sake of uniformity, we deleted “or spontaneously” from Figure 2.

p.15, l.319: “physiology”? *Sensu stricto*, physiology is a scientific branch. The authors might want to say rather: “physiological processes”.

Response: Corrected.

p.15, l.324: “nutrient intake”? perhaps better and more standardly: “uptake”;

Response: Corrected.

p.23, l.492: add volume and pages;

Response: Corrected.

p.23, l.497: Development (capital “D”);

Response: Corrected.

p.24, l.519: dynamicks? Correctly: dynamics;

Response: This is indeed the correct spelling as in the given article. Therefore, no changes required.

p.24, l.540: Correctly: PloS One;

Response: Corrected.

p.25, l.554: Nature Plants;

Response: Corrected.

p.25, l.559, p.27, l.612, p.28, l.623, p.28, l.637: correctly: of Sciences of the USA;

Response: Corrected.

p.27, l.605: Scientific Reports;

Response: Corrected.

p.28, l.620: the second parenthesis?

Response: Corrected.

p.28, l.628: volume??

Response: Corrected.

p.30, 31: Legends to Figs. 1 and 2: species names in italics.

Response: This error happened by copying the main text into the Figure legends mask in the browser. In the main manuscript, the species names are written in italics.

In conclusion, the paper needs only a very minor revision.

Response: We again thank the reviewer for their feedback.

Additional changes made by the authors

We fixed some referencing errors in the following sections and corrected the reference list accordingly:

- Example 1: Intrinsically powered triggering and motion in asynchronous snap traps
- Control over movement comes at the cost of additional energy requirements
- How costly are physiological processes?
- Complexity and diversity of trapping systems

and in the figure legends.

Additionally, the references for “*Cephalotus follicularis*” were updated in the SI file.